# Axial Orientation of Co-Crystalline Phases of Poly(2,6-Dimethyl-1,4-Phenylene)Oxide Films

**DOI:** 10.3390/polym12102394

**Published:** 2020-10-17

**Authors:** Manohar Golla, Baku Nagendra, Christophe Daniel, Paola Rizzo, Gaetano Guerra

**Affiliations:** Chemistry and Biology Department, University of Salerno, Via Giovanni Paolo II 132, 84084 Fisciano (SA), Italy; bakunagendra@gmail.com (B.N.); cdaniel@unisa.it (C.D.); prizzo@unisa.it (P.R.)

**Keywords:** *α* form, nanoporous-crystalline, WAXD, polarized FTIR, host-guest orientation, toluene, mesitylene, poly(2,6-dimethyl-1,4-phenylene)oxide, atactic polystyrene

## Abstract

Films exhibiting co-crystalline (CC) phases between a polymer host and low-molecular-mass guest molecules are relevant for many applications. As is usual for semi-crystalline polymers, axially oriented films can give relevant information on the crystalline structure, both by Wide Angle X-ray diffraction fiber patterns and by polarized Fourier-transform infrared spectroscopy. Axially oriented CC phases of poly(2,6-dimethyl-1,4-phenylene)oxide (PPO) with 1,3,5-trimethylbenzene (mesitylene) can be simply obtained by the stretching of CC PPO films. In fact, due to the plasticization effect of this highly boiling guest, PPO orientation can occur in a stretching temperature range (170–175 °C) nearly 50 °C lower than that generally needed for PPO films (220–230 °C). This low stretching temperature range allows avoidance of polymer oxidation, as well as formation of the mesomorphic dense *γ* PPO phase. Axially oriented CC phases of PPO with toluene, i.e., with a more volatile guest, can be instead obtained by the stretching (in the same low temperature range: 170–175 °C) of CC PPO blend films with polystyrene.

## 1. Introduction

Poly(2,6-dimethyl-1,4-phenylene) oxide [1,2,3,4], often abbreviated as PPO, is the only polymer beside syndiotactic polystyrene (s-PS) [5,6,7,8,9,10,11,12] that exhibits not only co-crystalline (CC) phases with low-molecular-mass guest molecules but also corresponding nanoporous-crystalline (NC) phases, as derived by guest removal [13,14,15,16,17,18,19].

As is well described in the literature, NC polymer samples can be useful for the removal of organic pollutants from water and air [20,21,22], in molecular sensors [18,23], in gas separation procedures [14,24,25], as support for catalysts [26,27] or even for chemical stabilization of unstable guest molecules [28].

CC polymer samples (mainly films) have been proposed for antimicrobial [29], optical [30,31,32], magnetic [33] and dielectric [34,35] applications. CC phases with active guest molecules can be obtained by guest sorption [36] or by guest exchange [37] from NC or CC phases, respectively.

As is well established for s-PS, polarized FTIR spectra of axially stretched films can be powerful tools to study the relative orientations of polymer chain axes and guest molecules in CC phases by evaluating the amount and sign of linear dichroism of guest peaks [28,33,38,39,40,41]. The dichroism of the FTIR peaks of guest molecules absorbed in oriented amorphous phases is instead negligible for both PPO [42,43] and sPS [44,45]. Hence, high dichroic ratios can be achieved only when guest molecules are prevailingly present in CC phases, rather than in the corresponding amorphous phases. This is easily achieved for s-PS due to the largely higher solubility and slower thermal removal of guest molecules in CC phases [38,39,40,41].

A prevalent presence of guest molecules in CC phases is, instead, not easily achieved for PPO films because they exhibit high guest solubility not only in their crystalline phases but also in their high-free-volume amorphous phases [13,14,15,16,17,18,19]. Moreover, axially oriented NC PPO phases cannot be obtained by the stretching of PPO films because the needed high temperatures (220–230 °C, i.e., close to *T_g_*) produce polymer degradation as well as the formation of the dense mesomorphic *γ* phase [42,43].

Axially oriented NC PPO phases were, however, obtained by stretching at lower temperatures of PPO/atactic polystyrene (aPS) blend films, thus exploiting the known plasticization effect of aPS on PPO [43]. For these blend films, the prevalence of guest molecules in the crystalline phase and hence the high guest dichroic ratios were achieved by guest sorption (benzene) from diluted (50 ppm) aqueous solutions [43]. However, this method is presently not available for pure PPO films and requires long sorption times, thus becoming unsuitable for bulky guests.

In this paper, we show that effective procedures to get axially oriented CC PPO phases can be based on axial stretching of CC films, rather than on sorption of guest molecules in axially stretched NC films.

## 2. Experimental Section

### 2.1. Materials and Film Preparation

Poly(2,6-dimethyl 1,4-phenylene)oxide (PPO), supplied by SABIC (Milan, Italy), has a weight average molecular weight (*Mw*) of 350 Kg/mol and a glass transition temperature (*T_g_*) of 220 °C. Atactic polystyrene (aPS), provided by Trinseo (STYRON 686E, Berwyn, Penn. USA) has *Mw* = 117 Kg/mol and *T_g_* = 105 °C. Solvents and guest molecules were supplied by Aldrich and were used without further purification.

PPO and PPO/aPS films, with thicknesses in the range of 70–120 μm, were obtained by casting procedures at room temperature from 3.5 wt% solutions in toluene or mesitylene. The obtained films exhibited CC PPO phases with toluene and mesitylene. NC films were obtained from CC films by treatment with acetonitrile at room temperature for 2 h, followed by acetonitrile desorption.

Film stretching experiments were conducted by a dynamometer INSTRON 4301(Norwood, MA, USA) with controllable stretching speed and temperature. The initial length between the dynamometer clamps (*l*_0_) and the initial width of the unstretched films were 20 mm and 10 mm, respectively. The film draw ratio λ = *l*/*l*_0_ (where *l* is the final length of the film) was in the range 2–5, using an elongation rate in the range 3–10 mm/min. Stretching experiments for CC PPO and PPO/aPS films were conducted at 170–175 °C.

Toluene sorption in NC PPO/aPS blend films was conducted at room temperature from a 50 ppm toluene solution in water.

### 2.2. Characterization Techniques

Two-dimensional (2D) wide-angle X-ray diffraction (WAXD) patterns were performed by a D8 QUEST Bruker diffractometer with nickel filtered CuKα radiation. Equatorial profiles were collected along the equatorial direction of the 2D patterns.

The degree of orientation of the crystalline polymer chain axes with respect to the stretching direction was evaluated using the Herman’s orientation function:(1)$$f_{c,\;WAXD}=\left(3\bar{cos^{2}x}-1\right)/2$$
by following the method described in detail in [43].

Fourier-transform infrared spectroscopy (FTIR) was conducted at 2.0 cm^−1^ resolution with a Vertex 70 Bruker spectrophotometer. It was equipped with a deuterated triglycine sulfate (DTGS) detector and Ge/KBr beam splitter. For polarized infrared spectra, this FTIR equipment was fitted with a SPECAC 12000 wire grid polarizer. The frequency scale was internally calibrated to 0.01 cm^-1^ using a He-Ne laser. Thirty-two scans were performed and averaged to reduce the noise level.

The degree of orientation of the crystalline host polymer chain axes with respect to the stretching direction was evaluated by the dichroism of the crystalline infrared peak located at 495 cm^−1^. The axial orientation factors *f*_c,IR_ were evaluated using the formula:(2)$$f_{c,IR}=\frac{\left(R-1\right)}{\left(R+2\right)}\frac{\left(2{cot}^{2}\theta +2\right)}{\left(2{cot}^{2}\theta -1\right)}$$
as described in detail in [43].

These *f*_c,WAXD_ and *f*_c,IR_ orientation factors are equal to 1 when all the chain axes of the crystalline phase are parallel to the stretching direction while equal to zero for random crystallite orientation.

Differential scanning calorimetry (DSC) scans were obtained by DSC Q2000 in heating ranges of 20–350 °C and a heating rate of 10 °C/min under continuous flow of nitrogen gas.

Thermogravimetric analyses (TGA) were carried out by TA instrument TGA Q500 (New Castle, DE, USA) with a heating rate of 10 °C/min in the temperature range of 20–600 °C under nitrogen gas flow.

## 3. Results and Discussion

Two well separated families of CC forms of PPO have been described: (i) highly ordered (exhibiting 4/1 PPO helices), only obtained with a few guest molecules (α-pinene, decalin, tetralin), which become amorphous after guest removal [46,47,48,49]; (ii) poorly ordered with highly extended polymer conformations, observed for many guests, which lead to NC phases by guest removal [13,14,15,16,17,18,19]. CC phases of family (ii) can assume two crystalline forms, which have been named α and β [19].

The experiments reported in this paper refer to the stretching of CC PPO phases with toluene and mesitylene, i.e., of two CC phases of family (ii) with α form.

### 3.1. Sorption of Toluene in Axially Oriented NC PPO Blend Films with aPS

In a recent paper, by using PPO/aPS (70/30; wt/wt) blend films, the first axially oriented CC form of PPO was obtained by benzene sorption from a 50ppm aqueous solution, in axially stretched NC α form films [43]. In this section, we describe an analogous sorption experiment for toluene. In particular, an axially oriented NC α form blend film (obtained by stretching at 190 °C up to a draw ratio of 3 for a PPO/aPS (70/30; wt/wt) blend film, exhibiting a WAXD pattern like that shown in Figure 2b,b’ of [43]), was immersed in a toluene 50 ppm aqueous solution for different times.

Polarized FTIR spectra of the NC axially stretched blend film after different sorption times are shown in Figure 1.

It is worth noting that toluene guest sorption, which is fast for NC PPO unoriented films, becomes slow for axially oriented blend films, even if NC. In fact, the guest vibrational peak at 729 cm^−1^ is barely detectable after four days of sorption and is well defined only after long-term sorption. For instance, after 25 days of sorption, the toluene content is close to 3.7% and 0.35 wt.% (as evaluated by TGA calibration of the absorbance of the FTIR 729 cm^−1^ peak) for the unstretched and stretched blend films, respectively. This indicates that the high permeability of the PPO amorphous phase is strongly reduced by axial stretching, possibly due to film densification associated with high temperature stretching. In fact, the density of the blend films of Figure 1 increases from 1.026 to 1.052 gm/cm^3^, as the draw ratio increases to 3. It is worth adding that an analogous densification phenomenon was already observed for fully amorphous PPO films [42].

The largely different intensities of the absorbance peaks for parallel and perpendicular polarizations indicate the presence of dichroism, not only for host peaks (indicated by h in Figure 1) but also for the guest peak (indicated by g in Figure 1), which is essentially independent of the guest sorption time.

The dichroism of the guest peak, more clearly apparent from the enlarged spectra of Figure 1b, is comparable with those of the polymer host peaks. In fact, the degrees of orientation, with respect to the stretching direction, of the polymer chain axes of the host crystalline phase and of the guest phenyl rings, evaluated on the basis of the dichroism of the isolated host crystalline peak at 495 cm^−1^ and of the guest peak at 729 cm^−1^, are *f_c,_*_IR,h_ = 0.45 and *f_c,_*_IR,g_ = 0.32, respectively. This clearly indicates that toluene, as already observed for benzene [43], when absorbed from aqueous dilute solutions, is preferably located as a guest of the cavities of the more oriented crystalline phase, rather than dissolved in the amorphous PPO/aPS phase. Moreover, this indicates that the mobility of the toluene guest molecule in the crystalline cavities of PPO is low, at least for the timescale of FTIR spectroscopy.

The sign of the dichroism of the 729 cm^−1^ peak indicates that the direction of the transition moment vector (TMV) associated with this C–H out-of-plane bending [50] is nearly perpendicular to the stretching direction. Since this TMV is also perpendicular to the aromatic ring, this indicates that (as already observed for benzene guest [43]) the guest phenyl rings in the crystalline cavity of the *α* form of PPO are preferentially parallel to the polymer chain axis.

Since this guest phenyl ring parallelism was already observed for the NC *ε* form of sPS, which has the crystalline empty space organized as channels parallel to the crystalline chain axes [8], the present results suggest that the empty space of the NC *α* form of PPO is possibly organized in a similar way.

### 3.2. Axial Stretching of Films with CC PPO/Toluene Phases

In this section, we describe two attempts to get films exhibiting axially oriented CC PPO/toluene phases.

#### 3.2.1. Axial Stretching of CC PPO/Toluene Films

A first attempt was based on the preparation of PPO/toluene *α* form films by casting from a polymer solution (3.5 wt.%) in toluene, followed by axial stretching.

TGA and DSC scans, at a heating rate of 10 °C/min, of this cast CC PPO/toluene film are shown as blue upper lines in Figure 2a,b, respectively.

The TGA scan of Figure 2a (upper curve) indicates a toluene content close to 11 wt.%, with guest removal starting at nearly 100 °C and ending close to 220 °C, i.e., close to the *T_g_* of the polymer. The DSC scan of Figure 2b (upper curve) shows a very broad endothermic peak corresponding to guest volatilization, which is superimposed on a minor melting peak roughly located at 270 °C. This melting temperature value is higher than those typical of NC PPO phases (in the range 247–255 °C) [16] and not far from those observed for DSC scans of other co-crystalline phases of PPO with highly boiling guests [51].

Stretching tests on these CC PPO/toluene *α* form films, in the temperature range from 190°C to 240 °C, give results very similar to those observed for NC PPO films [42]. In fact, it is possible to achieve crystalline phase orientation only for stretching temperatures close to the *T_g_* of PPO, i.e., in the temperature range 220–230 °C. These high stretching temperatures lead to two unwanted effects: (i) polymer oxidation [43] and (ii) formation of the dense (non-nanoporous) γ form [42]. Stretching tests at lower temperatures lead to film break for very low elongation values. This behavior is easily rationalized by the loss of most toluene molecules during the stretching tests in the used temperature range, which transforms CC PPO films to NC PPO films [16].

#### 3.2.2. Axial Stretching of PPO/aPS Films with CC PPO/Toluene Phase

A second attempt to obtain films exhibiting axially oriented CC PPO/toluene phases was based on the stretching of CC blend films as prepared by a toluene solution casting of PPO/aPS (70/30; wt/wt) blends.

The two-dimensional-wide-angle X-ray diffraction (2D- WAXD) pattern, as collected by sending X-ray beam parallel to the film surface (EDGE pattern) of the PPO/aPS (70/30; wt/wt) blend film, as cast from toluene solution, is shown in Figure 3a.

The 2D-WAXD EDGE pattern of the PPO/aPS (70/30; wt/wt) blend film, as cast from toluene solution, after partial toluene desorption at room temperature (toluene content ≈ 6wt.%) is shown in Figure 3a, and the corresponding equatorial profile is shown in Figure 3a’. These patterns show diffraction rings typical of the α form of PPO (at 2*θ*_CuKα_ ≈ 4.5°, 7.1°, 11.4°, 15.1°). These equatorial hk0 peaks are slightly polarized on the equator of the pattern, thus indicating the occurrence of a low degree of c_//_ orientation, i.e., of an orientation of the chain axes of the crystalline phase being preferentially parallel to the film plane [52].

Stretching of PPO/aPS blend films exhibiting CC PPO/toluene phase, rather than NC PPO phases, more easily leads to axially oriented CC PPO phases. In fact, the presence of toluene guest molecules has a plasticization effect, which adds to that of the aPS blend component. This leads to a downshift of the stretching temperature range (to 170–175 °C), which is definitely lower than the stretching temperature range for the NC PPO/aPS blends (185–195 °C) [43].

The 2D-WAXD EDGE pattern and the corresponding equatorial profile for the toluene cast blend film, after stretching at 175 °C up to a draw ratio of 3, are shown in Figure 3b,b’, respectively. As a consequence of stretching, the diffraction rings of the poorly oriented CC film (Figure 3a) are transformed in diffraction arcs (Figure 3b), thus showing that elongation induces axial orientation of the crystalline phase. The degree of axial orientation of the α crystalline phase, as determined by the azimuthal scanning of the (210) diffraction arc at 2*θ* = 11.4° for the EDGE pattern of Figure 3b, is *f_c_*_,WAXD_ = 0.47.

The stretching procedure at 175 °C also produces volatilization of most of the toluene guest molecules. In fact, the toluene content of the blend film reduces from 6 wt.% to nearly 0.2 wt.% after stretching.

Polarized FTIR spectra of the PPO/aPS blend film with CC PPO/toluene phase, after stretching at 175 °C up to a draw ratio of 3, are shown in Figure 4.

The degree of orientation of the polymer chain axes of the host crystalline phase of the stretched CC blend film, as evaluated on the basis of the dichroism of the isolated crystalline host peak at 495 cm^−1^, is *f_c,_*_IR,h_ = 0.55, i.e., higher than that measured for the stretched NC blend film (*f_c,_*_IR,h_ = 0.47 from Figure 1 and from [43]). The degree of orientation of the guest as evaluated on the basis of the dichroism of the guest peak at 729 cm^−1^ is *f_c,_*_IR,g_ = 0.49 and is close to that of the host and definitely higher than that measured for toluene guest molecules absorbed from dilute aqueous solution in stretched NC blend film (*f_c,_*_IR,g_ = 0.32 from Figure 1). This clearly indicates that residual toluene molecules, after stretching at 175 °C, are preferably located as guests of the cavities of the more oriented PPO crystalline phase, rather than dissolved in the amorphous PPO/aPS phase.

### 3.3. Axial Stretching of Films with CC PPO/Mesitylene Phases

TGA and DSC scans of the PPO CC films with mesitylene are shown as lower (brown) curves in Figure 2a,b, respectively.

The TGA scan of Figure 2a indicates a mesitylene content close to 16.4 wt%, with guest removal starting at nearly 100 °C and ending close to 280 °C. The DSC scan of Figure 2b shows two broad endothermic peaks corresponding to guest volatilization roughly centered at 135 °C and 210 °C, with the latter one superimposed on a minor melting peak roughly located at 270 °C, as is typical of CC PPO phases with non-volatile guests [51].

The volatilization temperatures of mesitylene from CC PPO films are definitely higher than those of toluene and are easily rationalized by the higher boiling point of mesitylene (*T_b_* = 164.7 °C) with respect to toluene (*T_b_* = 110.6 °C).

For PPO films exhibiting the CC PPO/mesitylene phase, axial orientation of the CC phase can be obtained by stretching at temperatures as low as 170–175 °C. i.e., at temperatures nearly 50 °C lower than those suitable for the stretching of NC PPO films or of CC PPO/toluene films (220–230 °C). This lower stretching temperature range is, of course, due to the plasticization effect of mesitylene, the content of which in the film after completion of the stretching procedure is still as high as 4.7 wt%.

Polarized FTIR spectra of the CC PPO/mesitylene films, after stretching at 175 °C up to a draw ratio of 3, are shown in Figure 5a. The vibrational peaks of the PPO chains of the host crystalline phase are clearly dichroic: 1143 (//), 1115 (┴), 828 (┴), 773 (//), 630 (//), 594 (┴), 563 (//), 518 (//), 495 (//), 474 (┴) and 414 cm^−1^ (//). The degree of orientation as evaluated on the dichroism of the isolated 495 cm^-1^ peak is *f_c,_*_IR_ = 0.40.

The spectra of Figure 5a also show the presence of an intense peak of the mesitylene guest at 688 cm^−1^, which exhibits a nearly negligible dichroism. This suggests that, after stretching, most of the mesitylene guest molecules are located in the poorly oriented amorphous phase.

Polarized FTIR spectra of the axially oriented NC PPO film, as obtained by guest removal by sorption/desorption of acetonitrile from CC PPO/mesitylene films, are shown in Figure 5b. The degree of orientation as evaluated on the dichroism of the isolated 495 cm^−1^ peak remains unaltered after guest removal (*f_c,_*_IR_ = 0.40).

Hence, axial stretching of CC PPO films with highly boiling guests allows, for the first time, the preparation of axially oriented NC PPO (unblended) films.

## 4. Conclusions

Axial stretching of CC and NC PPO films, due to the high polymer *T_g_*, has to be generally conducted in a high temperature range (220–230 °C), thus leading to polymer oxidation and the formation of a dense *γ* crystalline phase [42]. As a consequence, axially oriented CC and NC PPO phases have been till now obtained only for PPO/aPS blend films that, due to the well-known plasticizing effect of polystyrene on PPO, can be stretched at much lower temperatures (185–195 °C) [43].

In this paper, we show that axial stretching of PPO/aPS blend films exhibiting CC PPO/guest phases, rather than NC PPO phases, can more easily lead to axially oriented CC and NC PPO phases. In fact, the presence of guest molecules can have a plasticization effect, which can add to that of the aPS blend component. This leads to high elongations at break for stretching temperatures as low as 170–175 °C. The degree of crystalline phase orientation obtained by the stretching of the CC PPO/toluene blend films (*f_c,_*_IR_ = 0.55) is higher than that one obtained by the stretching of the NC PPO blend films (*f_c,_*_IR_ = 0.47). Moreover, stretching at 175 °C leaves residual amounts of toluene molecules, which are located essentially only in the crystalline phase.

In this paper we also show that, by axial stretching of CC PPO films with bulky highly boiling guest molecules (e.g., mesitylene), axially oriented CC and NC PPO films can be obtained without need for blending with aPS. For instance, the plasticization effect of mesitylene allows orientation of the crystalline phase (*f_c,_*_IR_ = 0.40) to be achieved by stretching at 175 °C in the absence of a polystyrene component. Moreover, axially oriented NC PPO films (*f_c,_*_IR_ = 0.40) are, for the first time, obtained by guest removal from axially stretched CC PPO films with mesitylene.

Additional stretching studies on CC PPO films with different highly boiling guests are in progress. Our aim is to obtain higher degrees of orientation, closer to those achieved by the stretching of sPS (*f_c,_*_IR_ even higher than 0.9). In fact, high degrees of axial orientation are helpful for host and guest structural characterizations, mainly by fiber diffraction spectra as well as by polarized infrared spectroscopy.

## Figures and Tables

**Figure 1 polymers-12-02394-f001:**
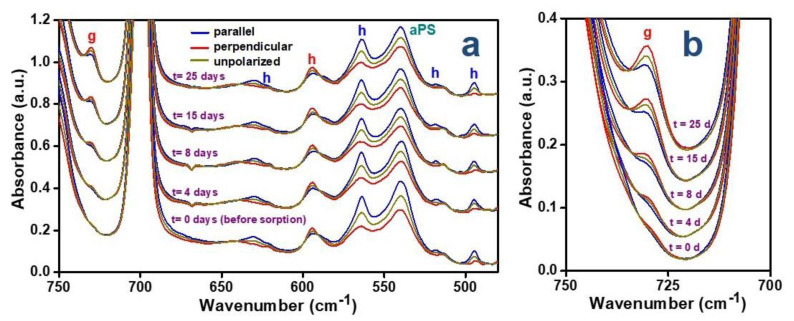
Polarized FTIR spectra of an axially oriented nanoporous-crystalline (NC) *α* form poly(2,6-dimethyl-1,4-phenylene)oxide (PPO)/atactic polystyrene (aPS) blend film collected with the polarization plane parallel (blue lines) or perpendicular (red lines) to the film stretching direction after different times of toluene sorption from a 50 ppm solution for two different spectral ranges: (**a**) 750–480 cm^−1^; (**b**) 750–700 cm^−1^. Labels (**h**) and (**g**) indicate peaks of the host PPO crystalline phase and of the toluene guest, respectively.

**Figure 2 polymers-12-02394-f002:**
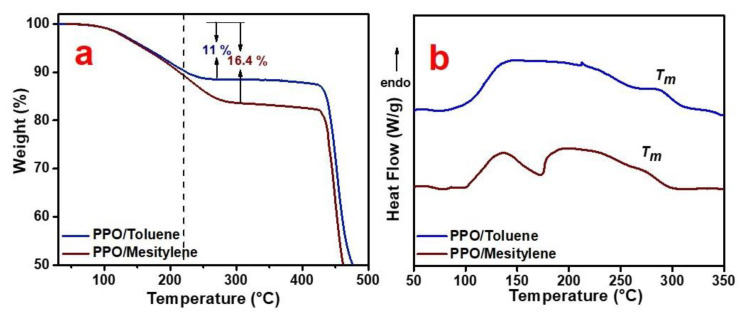
TGA (**a**) and differential scanning calorimetry (DSC) (**b**) scans at heating rate of 10 °C/min of PPO co-crystalline (CC) films with toluene (blue lines) and mesitylene (brown lines).

**Figure 3 polymers-12-02394-f003:**
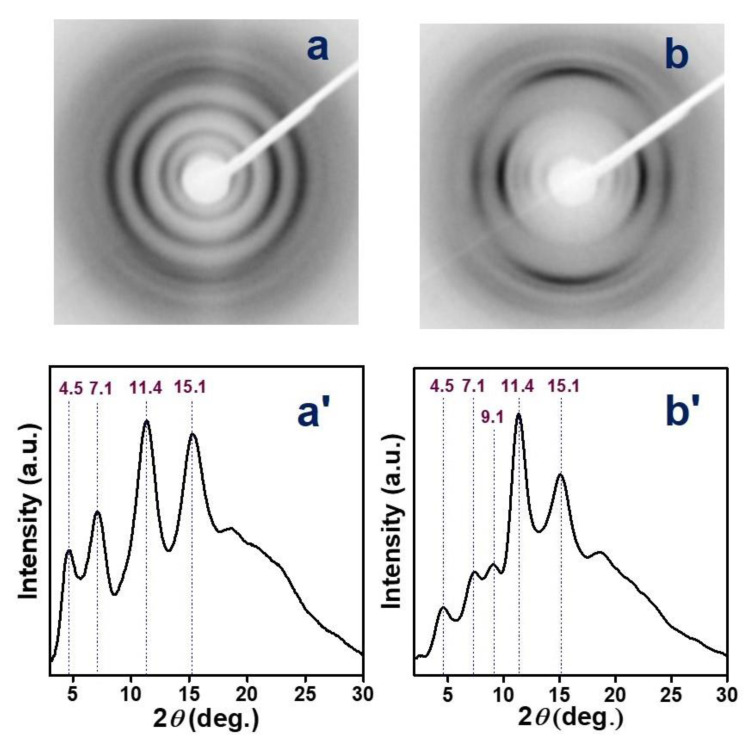
Two-dimensional-wide-angle X-ray diffraction (WAXD) EDGE patterns (**a**,**b**) and corresponding equatorial intensity profiles (**a’**,**b’**) of PPO/aPS (70/30; wt/wt) blend films as cast from toluene solutions: (**a**,**a’**) unstretched; (**b**,**b’**) stretched at 175 °C up to a draw ratio of 3.

**Figure 4 polymers-12-02394-f004:**
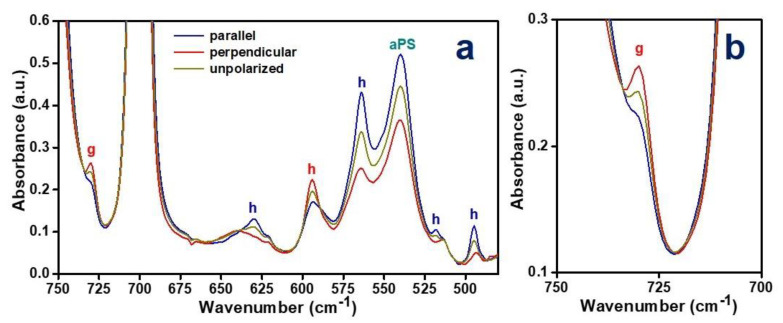
Polarized FTIR spectra of a PPO/aPS blend film exhibiting a CC PPO/toluene phase, after stretching at 175° up to a draw ratio of 3, as collected with the polarization plane parallel (blue lines) or perpendicular (red lines) to the film stretching direction: (**a**) 750–480 cm^−1^; (**b**) 750–700 cm^−1^. The unpolarized FTIR spectrum is shown as a grey line. Labels (**h**) and (**g**) indicate peaks of host PPO crystalline chains and of toluene guest, respectively.

**Figure 5 polymers-12-02394-f005:**
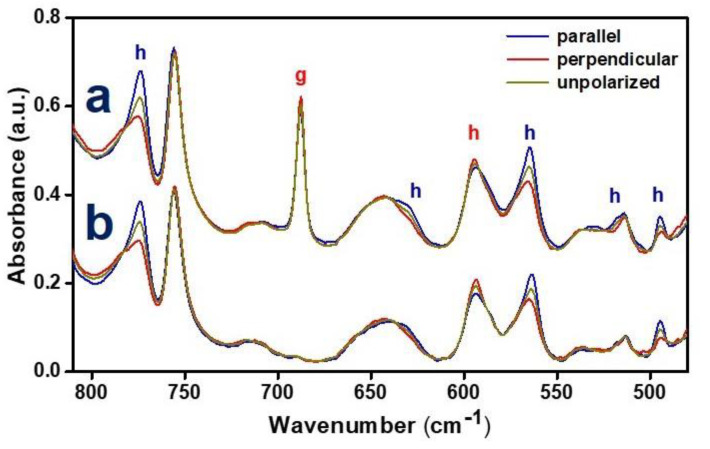
Polarized FTIR spectra of axial stretched PPO co-crystalline film at 175 °C up to a draw ratio of 3, (**a**) before and (**b**) after removal of the mesitylene guest by acetonitrile sorption/desorption.

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
