# Peer review of "Axial Orientation of Co-Crystalline Phases of Poly(2,6-Dimethyl-1,4-Phenylene)Oxide Films"

_polymers, 2020, doi:10.3390/polym12102394_

Round 1

Reviewer 1 Report

The author investigated the plasticizing effect of PPO by using mesitylene and blending with aPS with toluene. Alyhough the elongation of PPO has generally carried out at 220-230 C, the plasticizing allowed the elongation at lower temperature such as 170-175 C. These results are very interesting, so I reccomned to publish in the journal after considering the following comments.

  • aPS is one of the factors to reduce Tg of PPO. So I think that in “Key words”, it is better to add “aPS”.
  • Line 151: “oFilm” -> “of Film”
  • Line 205: The toluene reduces from 6 wt% to nearby 0.2 wt%. How was this reduction estimated? Is it DG?

Reviewer 2 Report

In this paper, the axial stretching of PPO/aPS blend films exhibiting CC PPO/guest phases, rather than NC PPO phases, has been studied. It is confirmed that axially oriented CC and NC PPO phases can be more easily produced. This phenomenon is related to the plasticization effect of the guest molecules as already reported in previous publications. Taking this into account, the significance of the present work is not obvious. Moreover, there are many problems of the manuscript, which leads to the manuscript cannot be accepted for publishing in Polymers.

  1. There is no evidence for the claim that “Dichroism of FTIR peaks of guest molecules absorbed in amorphous phases is generally negligible, due to conformational disorder and low degree of orientation of chains of the amorphous phase”. Actually, the different orientation status of polymer chains in the semicrystalline and amorphous phase can be characterized by using characteristic bands corresponding to these phase, respectively. It would much better to show these values for convincing the conclusion.
  2. The molecular weight dispersion of PPO and aPS should be given since smaller molecules in a sample with very broad molecular weight distribution may affect the discussed phenomenon of this paper as well.
  3. It is said that toluene guest sorption is fast for NC PPO unoriented films, while slow for axially oriented blend films. Why? This needs some explanation.
  4. From Figure 1, it is clear that the amount of absorbed toluene gets increased with time. How about the orientation of the absorbed toluene with different mass? In my opinion, it should not depend on the amount of the absorbed toluene.
  5. It is concluded that the degree of orientation of the polymer chain axes of the host crystalline phase and of the guest phenyl rings are fc,IR,h =0.45 and fc,IR,g=0.30, respectively. Does this value corresponds to all of the samples immersed in a toluene 50 ppm aqueous solution for different times? If not, the based IR spectra should be described clearly.
  6. The PPO/toluene α form films by casting from a 3.5 wt% solution in toluene contain 11 wt% toluene. However, the casting films of PPO/aPS toluene solution contain only about 6wt% toluene. What cause the big difference? Moreover, related TGA scan should at least given in the supporting information for comparison.
  7. It is said that stretching at 175 °C results in volatilization of the toluene guest molecules, leading to the toluene content of the blend film reduces from 6 wt% to nearly 0.2 wt% after stretching. However, in the corresponding polarized FTIR spectra, the g peaks of toluene guest can be clearly observed.
  8. There are also some English editing problems, such as more “clearly apparent”.

Round 2

Reviewer 2 Report

The manuscript has been carefully revised. It can be accepted.

Author Response

We have revised manuscript carefully.